# Exploring experiences engaging in exercise from the perspectives of women living with HIV: A qualitative study

Nora Sahel-Gozin[1], Mona Loutfy[1,2], Kelly K. O'Brien🆔[1,3,4]*

1 Institute of Health Policy, Management and Evaluation (IHPME), Dalla Lana School of Public Health, University of Toronto, Toronto, ON, Canada, 2 Department of Medicine, Temerty Faculty of Medicine, University of Toronto, Toronto, ON, Canada, 3 Department of Physical Therapy, Temerty Faculty of Medicine, University of Toronto, Toronto, ON, Canada, 4 Rehabilitation Sciences Institute (RSI), University of Toronto, Toronto, ON, Canada

* kelly.obrien@utoronto.ca

**Data Availability Statement:** All relevant data are within the manuscript and its Supporting Information files.

**Funding:** This research was supported by the Ontario HIV Treatment Network Endgame

## Abstract

### Objectives

To explore experiences engaging in exercise from the perspectives of women living with HIV, specifically, i) nature and extent of exercise, ii) components that characterize exercise experiences, iii) facilitators and barriers, and iv) strategies for uptake and sustainability of exercise.

### Design

Qualitative descriptive study involving online semi-structured interviews.

### Recruitment

We recruited women living with HIV from a specialty hospital, community-based organization, and medical clinic in Toronto, Canada.

### Participants

Ten women living with HIV who may or may not have engaged in exercise.

### Data collection

Using a semi-structured interview guide, we asked participants to describe their experiences with, facilitators and barriers to, and strategies to facilitate uptake of exercise. We electronically administered a demographic questionnaire to describe personal, HIV and physical activity characteristics of participants. We conducted a descriptive thematic analysis with the interview data, and descriptive analysis (medians, frequencies, percentages) of questionnaire responses.

Research Program – Breaking New Ground Award (EFP-1121-BNG) (https://www.ohtn.on.ca/). Kelly K. O'Brien is supported by a Canada Research Chair in Episodic Disability and Rehabilitation from the Canada Research Chairs Program (https://www.chairs-chaires.gc.ca/home-accueil-eng.aspx). The funders had no role in study design, data collection and analysis, decision to publish, or preparation of the manuscript.

**Competing interests:** The authors have declared that no competing interests exist.

## Results

Women characterized their experiences with exercise with six intersecting components: (1) culture, (2) gender, (3) HIV-related stigma, (4) episodic nature of HIV, (5) sense of belonging, and (6) perceptions of exercise. Facilitators to exercise included: aspirations to achieve a healthy lifestyle, using exercise as a mental diversion, having an exercise companion, and receiving financial support from community-based organizations to facilitate engagement. Barriers to exercise included: limited resources (lack of mental-health support and fitness resources in the community), financial limitations, time and gym restrictions, and cold winter weather conditions. Strategies to facilitate uptake of exercise included: creating social interactions, provision of online exercise classes, raising awareness and education about exercise, and offering practical support.

## Conclusions

Experiences with exercise among women living with HIV were characterized by intersecting personal and environmental contextual components. Results may help inform tailored implementation of exercise rehabilitation programs to enhance uptake of exercise and health outcomes among women living with HIV.

## Introduction

As the life expectancy of people with HIV rises, more individuals are living with the multidimensional health-related challenges of HIV, aging and multi-morbidity [1–8]. Together, these physical, mental, social, and cognitive health-related challenges may be experienced as fluctuating over time, termed episodic disability. Episodes of disability may be influenced by environmental and personal factors that can either improve or worsen disability over time [9, 10]. With HIV being a chronic illness characterized by episodic disability, there is a growing need for rehabilitation, and specifically exercise as a rehabilitation and self-management strategy with a potential to enhance health consequences among those aging with HIV [11–15].

Exercise can effectively improve health outcomes of adults living with HIV, including: body composition, muscular strength, cardiorespiratory fitness, functional capacity, mental health, and overall quality of life [16–22]. Despite the benefits of exercise, engagement in exercise varies. Systematic reviews and meta-analysis reported only 51% of people living with HIV achieving the recommended weekly physical activity guidelines of 150 minutes of moderate intensity physical activity [23], and that people living with HIV engaged in 533 minutes of sedentary behavior per day [24]. In particular, women living with HIV have lower exercise engagement compared to men living with HIV [25–28] and a higher likelihood of not achieving physical activity guidelines compared to their male counterparts [29]. While some literature exists on women's experiences living with HIV and exercise [25, 30–32], women comprise only 22% of the systematic review exercise literature [17], highlighting the need to better understand the experiences and perceptions of exercise among women living with HIV.

Our aim was to explore experiences engaging in exercise from the perspectives of women living with HIV. Specific objectives were to understand: 1) the nature and extent women living with HIV engage in exercise; 2) components that characterize exercise experiences; 3) facilitators and barriers to engaging in exercise; and 4) strategies for facilitating uptake and sustainability of exercise among women living with HIV.

## Materials and methods

### Study design

We conducted a qualitative descriptive study involving online semi-structured interviews with women living with HIV in Toronto, Canada [33]. This study was approved by the University of Toronto Health Sciences Research Ethics Board (Protocol #40852) (See S1 File). We obtained verbal consent for participation in the study, which was documented by the interviewer on the consent form prior to the interview.

### Patient and public involvement

This research involved a community-clinical-academic partnership involving women living with HIV, and HIV community organization and clinical settings. Specifically, we collaborated with an HIV community-based organization (AIDS Committee of Toronto), HIV specialty hospital (Casey House) and HIV clinic (Maple Leaf Medical Clinic) who facilitated recruitment of participants to the study. Furthermore, women living with HIV and community leaders in the field of HIV rehabilitation were involved throughout the study, specifically providing feedback on the interview guide and demographic questionnaire, engaging in a mock interview with the interviewer (NSG) to build skills and capacity of engagement with the study population, and facilitating knowledge translation opportunities of results with the HIV community.

### Participants, recruitment and sampling

We recruited women (cis, trans, and gender-diverse) living with HIV, 18 years of age or older, in Toronto, Canada with access to technology to participate in an online interview. We recruited participants from a specialty hospital (Casey House), community-based organization (AIDS Committee of Toronto), and a medical clinic (Maple Leaf Medical Clinic) in Toronto, Canada using a recruitment poster.

We used purposive and snowball sampling (word of mouth) to recruit a sample of women living with HIV who did and did not engage in exercise to obtain diversity of perspectives and experiences with exercise. We defined 'engagement in exercise' using the Canadian Society of Exercise Physiology (CSEP) physical activity guidelines as currently accruing (at the time of the interview) 'at least 150 minutes of moderate-to-vigorous-intensity aerobic physical activity per week' [34]. Women who did not meet this criterion were classified as 'non-exercisers.' Participants were categorized as an 'exerciser' or 'non-exerciser' using a combination of quantitative and qualitative approaches, including responses to the demographic questionnaire and their interview data. Participants who were categorized as non-exercisers based on their interview data were further categorized into one of the following three options: 1) currently does not exercise, 2) currently does exercise but does not meet the CSEP guidelines, or 3) currently does not exercise but was meeting the CSEP physical activity guidelines at some point in the past.

### Data collection

Interviews: The primary author (NSG), a female Master of Science candidate with experience in HIV and qualitative research conducted online semi-structured interviews using Zoom software for video communications [35, 36]. Only the interviewer (NSG), and participant were present at the time of the interview. We used a semi-structured interview guide to explore the following areas: 1) nature and extent of engaging in exercise (i.e., frequency and intensity of exercise history), 2) components that characterize exercise experiences (i.e., social and cultural

factors), 3) facilitators and barriers to engaging to exercise (i.e., practical considerations), and 4) strategies for future and sustained exercise engagement (i.e., ways in which to encourage women to uptake exercise) (See S2 File). Field notes were also taken during the interviews. All interviews were audio-recorded and later transcribed verbatim.

Demographic Questionnaire: After the interview, we electronically administered a demographic questionnaire using Qualtrics XM [37] to describe personal, HIV, health and exercise characteristics of the sample.

Participants received a $30 CAD electronic gift card as a token of appreciation for their participation in the study.

## Data analysis

**Interview data.** We analyzed transcripts using a descriptive thematic analysis informed by Braun and Clarke to classify themes within the data [38]. The interviewer (NSG) transcribed the interview audio files verbatim and reviewed each for accuracy. The interviewer (NSG) created preliminary notes, and drafted participant summaries for each interview embedded with participant's characteristics and experiences with exercise. The interviewer (NSG) coded all transcripts line-by-line to establish a preliminary coding scheme that pertained to the study objectives. Co-authors (KKO, ML) reviewed a sub-sample of the transcripts and participant summaries. We defined and clustered the codes into broader categories and sub-categories. All authors met on 4 occasions to refine the coding scheme and discuss analytical interpretations.

**Questionnaire data.** We downloaded responses from the demographic questionnaire from Qualtrics into Microsoft Excel 2018 [39]. We conducted a descriptive analysis including frequencies (percent) for categorical variables and median (25th,75th percentiles) for continuous variables to describe the characteristics of the participants.

## Sample size

We aimed to recruit an estimated sample size of 12–15 participants to gain the perspectives of women living with HIV and provide depth for rich data analysis [40]. Past qualitative studies that explored experiences with exercise among adults living with HIV were able to address study objectives using a similar sample size [41–43].

## Results

Of 16 women who showed interest in the study, 10 were eligible, agreed to, and participated in the study. Each interview was approximately 60–90 minutes in duration. See Table 1 for the characteristics of participants.

## Exercise engagement as determined by the self-reported questionnaire

Exercise characteristics of participants as reported in the self-reported questionnaire are displayed in Table 2.

## Exercise engagement as determined by the interview data

Seven women were classified as non-exercisers and three as exercisers based on interview data as per the CSEP guidelines as "currently accruing (at the time of the interview) at least 150 minutes of moderate-to-vigorous-intensity aerobic physical activity in the past week" [34]. Of the seven non-exercisers, three currently exercised but did not meet the CSEP guidelines, two

**Table 1. Characteristics of participants (n = 10).**

| Personal Characteristics | Median (25th, 75th percentile) |
|---|---|
| Age in years (n = 10) * | 54 (49, 57) |
| Personal Characteristics | Number (%) |
| Race and/or Ethnic Background (n = 10) * | |
| Black or African (e.g., American, Canadian, Caribbean) | 4 (40%) |
| White (e.g., European) | 2 (20%) |
| Indigenous (e.g., Anishinaabe, Inuit, Iroquois) | 2 (20%) |
| White and Indigenous | 1 (10%) |
| South Asian (e.g., Indian, Pakistani, Punjabi, Sri-Lankan) | 1 (10%) |
| Relationship Status (n = 9) | |
| Single | 7 (70%) |
| Married | 1 (10%) |
| Prefer not to answer | 1 (10%) |
| Has Children (n = 10) * | 6 (60%) |
| Live Alone (n = 9) | 4 (40%) |
| Highest Level of Education Completed (n = 9) | |
| Did not complete high school | 1 (10%) |
| Completed high school | 1 (10%) |
| Completed College | 4 (40%) |
| Uncompleted University | 1 (10%) |
| Completed University | 2 (20%) |
| Average Yearly Gross Income (n = 9) | |
| Less than $20,000 CAD | 3 (30%) |
| $20,000 to $40,000 CAD | 4 (40%) |
| More than $40,000 CAD | 2 (20%) |
| Main Source of Income (n = 9) | |
| Full-time employment | 2 (20%) |
| Part-time employment | 2 (20%) |
| Unemployed | 1 (10%) |
| Ontario Disability Support Program (ODSP) | 3 (30%) |
| Canada Pension Plan (CPP) | 1 (10%) |
| HIV and Health Characteristics | Median (25th, 75th Percentile) |
| Year of HIV Diagnosis (n = 10) * | 1999 (1993, 2010) |
| HIV and Health Characteristics | Number (%) |
| Taking anti-retroviral medications (n = 9) | 9 (90%) |
| Undetectable viral load (<50 copies/mL) (n = 9) | 9 (90%) |
| Smoking History (n = 9) | |
| Never smoked | 4 (40%) |
| Smokes occasionally (in the last 30 days) | 1 (10%) |
| Currently smokes (in the last 30 days) | 3 (30%) |
| Former smoker (have not smoked in the last 30 days) | 1 (10%) |
| Living with at least 1 other concurrent health condition in addition to HIV (n = 9) | 8 (80%) |
| Concurrent Health Conditions (experienced by $\geq$ 2 participants) (n = 9) | |
| Mental health condition (e.g., depression, anxiety) | 5 (50%) |
| Lung disease (such as asthma, chronic bronchitis, emphysema) | 4 (40%) |
| Gastrointestinal conditions (e.g., stomach ulcers and diarrhea) | 3 (30%) |
| High cholesterol (elevated levels of cholesterol in the blood) | 3 (30%) |
| Addiction or substance use disorder (e.g., alcohol, drugs) | 2 (20%) |

(*Continued*)

**Table 1.** (Continued)

| | |
|---|---|
| Chronic pain—joint pain (e.g., arthritis) | 2 (20%) |
| Cognitive decline (e.g., memory loss, confusion, trouble thinking clearly) | 2 (20%) |
| Osteopenia or osteoporosis (e.g., decreased bone density) | 2 (20%) |
| Trouble sleeping (e.g., insomnia) | 2 (20%) |
| Self-reported current health status (n = 9) | |
| Excellent | 1 (10%) |
| Very good | 1 (10%) |
| Good | 6 (60%) |
| Poor | 1 (10%) |

*Note. One participant did not complete the demographic questionnaire; some characteristic responses to the demographic questionnaire were supplemented by this participant's interview data indicated by characteristics (n = 10).

**Table 2. Self-reported level of physical activity (n = 9).**

| Physical Activity Characteristic | Number (%) |
|---|---|
| In this past week (7 days), did you engage in at least 150–300 min of moderate-intensity aerobic physical activity, or at least 75–150min of vigorous-intensity aerobic physical activity, or an equivalent combination of moderate-intensity and vigorous-intensity activity? | |
| Yes (defined as exerciser) | 5 (50%) |
| No (defined as non-exerciser) | 4 (40%) |
| Did you engage in muscle or bone strengthening activities (resistance activities) at moderate or greater intensity at least 2 days this past week (past 7 days) (e.g., lifting heavy objects, pushups, cycling, heavy gardening)? | |
| Yes | 5 (50%) |
| In the past two months I limited the amount of time spent being sedentary, meaning I replaced sedentary time with some form of physical activity of any intensity (including light intensity) which provides health benefit. | |
| Strongly Agree | 1 (10%) |
| Agree | 4 (40%) |
| Disagree | 3 (30%) |
| Strongly disagree | 1 (10%) |
| Did you engage in functional balance and strength training at moderate or greater intensity on 3 or more days this past week (past 7 days)? | |
| Yes | 4 (40%) |
| Has your exercise changed during the COVID-19 pandemic? | |
| Yes, I exercise more during the COVID-19 pandemic | 2 (20%) |
| Yes, I exercise less during the COVID-19 pandemic | 6 (60%) |
| No, my exercise has not changed | 1 (10%) |
| Which statement best describes you? | |
| I currently do not exercise, but I am thinking about starting to exercise in the next 6 months | 2 (20%) |
| I currently exercise, but not regularly | 2 (20%) |
| I currently exercise regularly, and have done so for longer than 6 months | 4 (40%) |
| I have exercised regularly in the past, but I am not doing so currently | 1 (10%) |
| Does exercise play an important role in your life? | |
| Yes | 8 (80%) |

Note. Nine (9) out of the ten (10) study participants completed the demographic questionnaire.

currently did not exercise but was meeting the CSEP guidelines at some point in the past, and two currently did not exercise.

## Type, frequency and duration of exercise

Based on interview data, the type, frequency and duration of exercise engagement varied among the women living with HIV. Across all 10 participants, the most common type of exercise activity performed in the past week was walking (n = 6), and the least common was calisthenics and kick boxing (n = 1). Other activities included: cycling (n = 4) and group-based exercise like Zumba and yoga (n = 4). Frequency of exercise ranged from women who had not exercised in over 5 years (n = 1) to women who exercised every day of the week (n = 2). The duration of time the women exercised ranged from no exercise (n = 1) to seven or more hours a week (n = 2). See S3 File for further details of exercise frequency, duration and type.

## Characterization of experiences with exercise

Six intersecting components collectively characterized and constructed experiences with exercise among women living with HIV: (1) culture, (2) gender, (3) HIV-related stigma, (4) episodic nature of HIV, (5) sense of belonging, and (6) perceptions of exercise (Fig 1). We describe the components that comprised exercise experiences with representative quotes below.

**Culture.** Women described culture as influencing their attitudes, values and behaviour to exercise. Culture was comprised of i) cultural expectations, ii) intersectionality of women and their social identities, iii) exposure to exercise, and iv) food diversity.

*i) Cultural expectations.* Participants spoke about the cultural expectations, which influenced their engagement with exercise. Expectations of women were associated with specific responsibilities and customs, specifically those of African and/or Black decent. Participants whom identified as Black or African expressed that their cultural background largely dictated how women engaged and perceived exercise, namely with prioritizing domestic and other cultural responsibilities over their own health. Regardless of living in Canada, one community member expressed that ethnic women may be socially inclined into respecting their native culture, which places boundaries on how exercise is experienced:

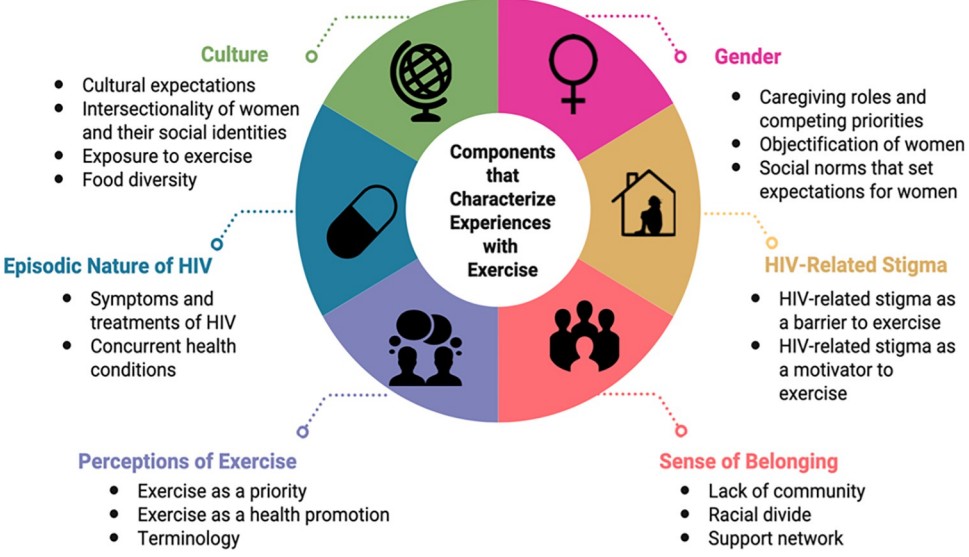

**Fig 1. Components that characterize experiences with exercise among women living with HIV.**

*Culture, you know, as African woman, there is something I can do, there is something I cannot do because of my culture. To respect also where I come from. . . we don't talk about sometime because we are in Canada, we wanted to get this Canadian system, but even in that, there is something I can do, there is something I cannot do. (P2)*

This participant further described cultural expectations that required dressing modestly when entering fitness spaces:

*Even I go to YMCA, some they put like uh, what, the short-short uh to go to do the gym. . . But, me for my culture, no, we put the long one. . . to cover my leg. . . And there is uh something to cover my-my-my stomach, to cover-not to show my body. . . because of what? Because of our culture. (P2)*

Cultural expectations also included the need to accommodate others by fulfilling domestic responsibilities from cooking to cleaning, leaving no time to participate in exercise:

*Especially in the African/Caribbean communities. . . Yes, yes [self-care] is the last [priority]. . . because culturally, a woman with her child, there is uh something, she cannot do like exercises, she there just to cook, to go to bring the children to school, to wash clothes, to do that. (P2)*

Another participant spoke about putting her nephew's needs before her own, limiting her engagement with exercise:

*I live with my nephew who is new to Canada. So, he still has some culture from back home. . . last night, I had, uh, prepared everything to do my back exercise but there was no food so I had to make sure I cook and I spend, like uh, 3 hours in the kitchen. So, by the time I finished, I didn't want to exercise. . . Men can ignore [chores] but women cannot. (P1)*

*ii) Intersectionality of women and their social identities.* A couple of women expressed that exercising was not important for some women of color who experienced intersectionality, where they faced multiple forms of inequality. Having dealt with challenges such as oppression from others as a racial minority, to experiencing stigma due to their HIV status, exercise as a result, fell to the bottom of some women's priority list:

*I find African women are affected differently than European women based on oppression. There's just more intersectionality's with women of colour. We have to deal with oppression. . . in Canada for women of colour going to the gym is not going to be their priority. It's not. Especially, if they're dealing with health, and children, and poverty, and racism and, maybe hiding that they're gay, or hiding their diagnosis from their family. Like, it just layers up, right? (P3)*

Another participant of ethnic minority also spoke about intersectionality in women living with HIV, and how it may limit their exercise experience:

*In women it's so many other issues. . .the cultural issues, the children, the financial, this-that, the other taboos of allowing-not allowing. Many things play-play a factor. . . (P9)*

*iii) Exposure to exercise.* Living in Canada also allowed one woman to become exposed to the benefits of exercise, where she was once not privileged in her native country. Having

experienced that exercise can be a healthy self-management strategy, she was better able to understand the benefits of exercise, as demonstrated below:

*with education now I never talked about it like long time ago. . .also because of the culture or the exposure to know those benefits. . . let's say when I was still in my home country, but uh, since I've been in Canada, being exposed it to how exercising will uh, have uh, a positive impact on my immune system, my health. (P1)*

With readily access to internet in Canada as opposed to her home country, another participant of African-Black decent agreed that offering online exercise classes would be beneficial for women:

*Yeah, especially here in Canada 'cuz most people have access to internet. It's not like back home, so that-that would be easier. (P5)*

*iv) Food diversity*. Additionally, culture produces diversity in food, with one participant claiming that many ethnic women engage in unhealthy eating habits that are transferred to their children. This should further coerce women into reconsidering their relationship with food and consult one another to gain insight into portioning meals alongside working out to ultimately lead a healthy life:

*And a lot of women too, we're all ethnic a lot of us have diversity in terms of food eating. That's a large, huge-huge barrier for women who are overweight because we come from [inaudible] were if the plates not full of food, you're offended. . .. [Women]have to see how they look at food because not only would it benefit them, it would benefit their children. Women of color are bringing up children that are obese as well. Right, so if they're working out and they know how to portion food for them properly, and they see their friends doing it the same way and they're saying it to themselves. You know, Nigerian women are talking to themselves in Nigeria or women from Turkey are talking to themselves in Turkey and telling themselves, "no, do it this way sis, it really does [inaudible], and your kids will like it too and do smaller portions. . . (P3)*

Another participant of ethnic minority, while not directly driven by culture, also conferred the importance of maintaining a healthy diet alongside exercise:

*Exercise also, that's why I told you, it go together with diet. . ..uh diet and exercise and the treatment because the treatment, diet, the food that we eat is a treatment too. Yeah, treatment. The food we eat is uh treatment for our body. (P2)*

**Gender.** Identifying as a woman, affected the ways in which some participants viewed and engaged with exercise. Gender was comprised of i) caregiving roles and competing priorities, ii) objectification of women, and iii) social norms that set expectations for women.

*i) Caregiving roles and competing priorities*. For some participants, competing priorities at some point in their lives, such as work or caregiving (i.e., children) took precedence over exercising:

*There was a time in my life where I had to prioritize my kids. . .'cuz they were little. So, there's no way, I was going to say, "you know what, let me forget—let me just focus on the exercise. I*

*just need to focus on them." With me, I think the stages in life, that's how I look at it. . .every-every person has stages in life. . . I think there was a time I almost gave up on exercise all together because it wasn't just working, I was exhausted. You have a full-time job and you have small kids, you're attending diapers, they're crying, they're not well and you have to cook. (P5)*

This was similarly expressed by another participant who articulated that some women struggle to engage in exercise because of gendered factors:

*Well, women with children, yeah, it's difficult for them to uh be able to get the time [to exercise].(P10).*

Another woman stressed that she was a full-time caregiver to her elderly parents, further limiting her time to exercise:

*[my current position] stops me, limits me [to exercise] at this stage of my life because now I'm a caregiver full-time with two-two babies. Yeah, to mama and papa. (P9).*

Furthermore, maintaining a working life, while also exercising proved difficult for another participant:

*I also have a couple of jobs so I have to work around that as well, right? . . . I have to work around my-my life, my daily life as well. My responsibilities and stuff, I have to work around that, and my obligations I have. (P6)*

*ii) Objectification of women.* The objectification of women was viewed as the sexualization of women in fitness spaces and at large. One participant voiced her concern over the objectification of women's bodies in the context of weight-lifting:

*Well, especially with weights because I find there's a lot more- there's more girls now, but there's a lot of guys too. So, yeah in that sense, I don't like being looked at as a piece of meat . . .and that happens a lot. . . just getting that look or those cat calls and that kind of crap, I just could knock somebody out. (P8)*

Consequently, exercise was viewed as a source of strength, and a means for some women to defend themselves:

*you get so strong, you can beat up those-those guys that are getting in the way or bugging down your life. (P8)*

Similarly, another woman of racial minority also voiced that exercise gave her the power to defend herself against the threat of misogyny:

*As a woman, I'm 40% weaker than men and a lot of us women of color have been abused by misogyny for far too long. And, when you exercise, you build strength where you can take a weapon and balance it out. And, there's women right now as we speak in India that are learning how to fight with sticks and swords. (P3)*

*iii) Social norms that set expectations for women.* Participants stated that social norms set the tone for body image and beauty standards for women, and consequently encouraged them

to engage in exercise to look a certain way. This influenced engagement in exercise as some women claimed to rely on social media to obtain their ideal body standards in hopes of looking like other women who post exercise material:

*I'm watching this Instagram; I see some people doing exercise. . . So, I want to look like them, like this is the motivation. (P7)*

This was echoed by another participant about other women in the HIV community:

*So, in terms of what motivates me [to exercise], is myself 'cuz I don't do social media. . . people in the community of what we're discussing. Uh, everyone's social media based and that's where they get their culture, that's where they get their influence, that's where they get their motivation, right? (P3)*

Some participants also described how body image, specifically, 'looking good' was important for most women and encouraged them to exercise. For instance, one participant indicated:

*I like to look good. I like when I dress, I look good and I don't like too much fat. You know, and yeah, I like to, you know, fit good in clothes and be happy with what I see in the mirror. . . us women like to look good. (P5)*

Similarly, another participant confirmed that gender influenced their level of engagement with exercise:

"Well, being a woman, I don't know, we have to look good, right?" (P10)

The social norms geared towards women may also demand a traditionally 'feminized' approach to working out. This participant spoke on the negative stereotypes of women in body-building:

*Well, the-the-the social norms and bias about gender, about women working out, looking beastly, looking too manly, it's too unfeminine, it's unsexy, you know. So, it's always characterized, especially as looking beastly, you know. And-and-and unattractive. So, I've had to break through those norms and stereotypes. (P3)*

**HIV-related stigma.**   Stigma, defined as negative beliefs and attitudes about people living with HIV, influenced participants' decisions to engage in exercise. For some women, issues related to stigma generated fears which resulted in i) HIV-related stigma as a barrier to exercise, whereas for one woman, it resulted in ii) HIV-related stigma as a motivator to exercise, as a means to alleviate frustration from stigma.

*i) HIV-related stigma as a barrier to exercise*. Both social and internal stigma around HIV emerged as a challenge, with some women voicing concerns for those who may fear exercising due to their HIV status:

*Stigma is very critical in our situations, right? Especially, if it's known to others, right? So, and people can be very mean. . . Like, uh even catching [HIV] from kissing, all these myths that they've created through the years. It's just crazy, right? . . . Some women yeah, because they're*

*afraid. I've seen some people; they get afraid because they sweat and all that stuff. And, people think they're going to catch [HIV]. . . The dirty looks, the rudeness, the ignorance as a whole. (P6)*

One woman noted that some of her fellow community members avoided exercise programs related to living with HIV due to stigma:

*To be honest, if this uh program, you send it, for me I don't-I don't mind. For other, all they see is it will help you because you are HIV-positive. Just that word, just that word, finish! It goes inside for that person and say "no. . .maybe this person I am living with will know what is happen to me or if I go-mm mm I don't want." Stigma is uh something that make people to be dis-, it make people to die. (P2)*

Another participant echoed this concern, affirming that women living with HIV are still stigmatized:

*I can see why it might be a little bit terrifying for people in my situation to put themselves out there because it's so stigmatized for what I'm living with. . . And, even just getting fit, a lot of us feel, you know, "where do I begin? What do I do? How would that look like?" (P3)*

An HIV diagnosis also meant that many women experienced unfavorable symptoms that came with their medical treatment, such as lipodystrophy or atrophy:

*I don't know if it's medication or HIV itself- uh, some unfortunately, can uh, have like uh, should I call it a deformed body or. . ..? That-that sounds negative. Uh, uh, fat disproportion, like where we find someone with a big chest or uh, or uh big uh, shoulder while their lower part is uh, like muscles are losing on the lower part is losing, kind of extremities or this part is losing muscles things like that, so exercise can help maintain muscles or uh, or uh, yeah prevent it to go bad. (P1)*

*ii) HIV-related stigma as a motivator to exercise.* While participants expressed the negative effects of stigmatization towards women living HIV and its discouraging impact on exercise engagement, one participant also voiced that exercise can be employed to offset the negative effects of stigma, facilitating her engagement in exercise:

*[Exercise] helps [women] deal with stigma and everything else that might be happening in their lives, you know? (P6)*

**Episodic nature of HIV.** Women living with HIV revealed that they experienced episodic disability, characterized by fluctuating periods of feeling healthy and unwell which influenced their experiences with exercise. Living with HIV was viewed as having its own set of consequences related to i) symptoms and treatments that occasionally resulted in difficulties when trying to perform daily tasks. A majority of the women also revealed that in addition to living with HIV, ii) they were living with other concurrent health conditions, such as diabetes and COPD, which further contributed to their episodic disability.

*i) Symptoms and treatments of HIV*. Despite the women actively making efforts to take their daily HIV medication and maintain their health, adverse effects occasionally took a toll on some of their physical and mental well-being. For instance:

*there's a huge correlation between the medication they put us on and depression. (P3)*

Because of this, some women faced difficulty finding motivation to exercise. For example:

*Anything that discourages me [from exercising]? Uh, I don't know, just I guess, my-my uh like depression. . . just feeling down . . . being HIV, you know. . . it's depressing. (P10)*

HIV and medication also caused a shift in their body, specifically with weight. For instance, experiencing lipodystrophy left this woman limited with how much she engaged with exercise:

*I can do a 2.5 hour walk, nice hike, consistent pace, but not running. My body can't do it. . . I'm like pineapple on popsicle sticks, where my body's kind of deformed with over lipodystrophy and years of whatever. And uh, but my legs are really thin, my ankles are really tiny just like my—but, anyways, the lipodystrophy. . .So, all the time those popsicle sticks crack and I would fall because of my ankles. (P9)*

Other women spoke about the importance of exercise for the health of adults living with HIV and the effects of lipodystrophy. For example:

*[Exercise] is very important because the medication, first of all, they make you gain weight in the wrong places. . . so if you're not careful and you're just eating and eating, you find yourself with a big stomach and your bum is small. . . already that-that's not proportional, like it's disproportional. . . [the medication] causes that. It's called lipodystrophy or something like that. (P5)*

One participant noted that their medication resulted in fatigue, which sometimes affected their ability to exercise:

*Because of my medication, I get tired and uh so I might not have a good day with exercise. (P6)*

Another participant highlighted the uncertainty associated to HIV and aging. Lack of knowledge had produced a degree of uncertainty living with HIV, which influenced engagement in exercise:

*As we get older, there's not a whole lot of studies. I mean, this disease has only been around for. . . 50–60 years since the cases. . . We don't know how all the men are aging, they're just starting to age now, let alone women . . . so I got to make sure I got some sort of regime in place now while I can before-before it gets the better of me. (P4)*

*ii) Concurrent health conditions.* The majority of the women in this study were living with other chronic health conditions in addition to HIV. Living with multi-morbidity interrupted certain participants from engaging in exercise as it put strain on their bodies, which further had the potential to exacerbate episodic disability living with HIV. For instance, one participant noted:

*I've had a lobectomy in my left lung, I had lung cancer so I have respiratory problems, I get a lot of pneumonia and I also have lymphedema . . .it's just really hard for me to breath and aerobics it's just too much. . . as recently I've even gotten worse because now, I've had an*

*exacerbation. . . of COPD. . . I've had lymphedema uh so, uh and that's just impossible to breath. It's impossible to walk from like 3 feet. (P8)*

Similarly, another woman described her experience living with asthma and how its limited her exercise:

*I have uh negative exercise because uh, sometime, because of my condition. . . I'm like asthmatic. So, I needed to, at that, I don't need to do more than as usual-as I'm supposed to do for my health because if I do that, maybe that problem will come back. (P2)*

Other chronic conditions, such as diabetes, also affected exercise engagement among women in the study:

*[I exercise] depend[-ing] on how I feel when I wake because I also have diabetes as well. . . (P6)*

**Sense of belonging.** Having a sense of belonging within the HIV community, or lack thereof, could encourage or deter a woman from engaging in exercise. Women expressed the importance of finding a sense of belonging within a community when dealing with the challenges related to HIV. Sense of belonging was comprised of i) a lack of community that discouraged women to exercise, ii) a racial divide within certain spaces, and iii) having a support network that garnered motivation to exercise.

*i) Lack of community.* One participant agreed that being part of a group would motivate her to exercise, but she struggled to find a community group within the multiple sub-groups of people living with HIV:

*Even looking for a support group, again, everything is male-based. . . racially I don't fit in the demographics of people living with AIDS as women. So, it's even harder, 'cuz then they get segregated down by their races. . . But, then even the women that I've met, there's subgroups for them. But, there's not someone that's like, you know, old ladies who are white that, you know, get together and exercise. . .I just don't feel comfortable bombing their like group or whatever. So, it's hard to find that right group. . . (P4)*

Another woman described her experiences as a long-time survivor of HIV as 'othered' due to her identity:

*A lot of women like myself, won't go to these socials because we're the odd man out . . . a lot of women that feel very uh othered because of it. . . it's only a certain type of woman that will access these programs . . . women that are been diagnosed for 20–30 years are not accessing anything, anymore . . .Like myself, I don't access any of these organizations. . . there's no contingencies for women like us, there's no support for women like us. (P3)*

Similarly, this participant was discouraged from continuing her exercise group (non-HIV related) as she felt unwelcomed and overwhelmed by the intensity level:

*I managed to get into that meet up group for hiking and walking in my area. . . as soon as they started, they were like 500 million miles ahead of me. I couldn't keep up. . . Inconsiderate to the person that I'm new. . . you didn't even care to see if I existed or not. So, I don't know if it was a racial thing or if I was just paranoid or whatever. But, as soon as I noticed that*

*happening before the trail got too far in, I decided a smart thing is to turn around. . .that was not a good experience so I never went again for a meet up. (P9)*

*ii) Racial divide.* One participant, who represents a racial minority, described a sense of intolerance she experienced within the gym space. This resulted in her to feel discomfort when trying to engage in exercise:

*You know, going to the gym, it's 90% Caucasian. You know, 40%-30% women, 60% men, you know? Even going to those environments, people look at you like "what are you doing here?" Like, Toronto's racist as hell. And so, when you're stepping out of your comfort zone and you get these micro-aggressions, working out, which should be a priority, becomes an invasion of privacy because of these micro-aggressions. (P3)*

This was echoed by another participant:

*The color of the skin, the some of the people, sorry to say this, the color of the skin, "oh I don't want to be the part of this because it's just, I see uh black women, some black men. I don't want to be part of that. " . . . "I don't want to go to gym, uh with-with black people, maybe- I don't know" there are some people like that. (P2)*

*iii) Support network.* Conversely, some women claimed they found a support network (i.e., fitness instructor and/or exercise class) who encouraged them to get involved and remain consistent with exercise:

*by yourself, it's hard [to exercise consistently]. But, with a teacher, that is the reality. With a teacher, you go there and let's go! Put the music and everyone is doing and there is different level, but your power, you have to do it. (P7)*

The role of support as promoting engagement in exercise was reiterated by another participant:

*I maintain that one-day a week [of group-based exercise] 'cuz I know that there's somebody else that I can go with and things like that, it's accountability. (P4)*

**Perceptions of exercise.** Each participant had different perceptions of exercise, which influenced the role it played in their life. Perceptions of exercise was comprised of i) exercise as a priority, ii) exercise as health promotion that could prevent secondary disease with aging and multi-morbidity, and iii) understanding the terminology between 'physical activity' and 'exercise' can promote exercise among women living with HIV.

*i) Exercise as a priority.* Some participants expressed exercise as a luxury (i.e., financial, time, energy) rather than a priority, which largely became non-attainable to them. This meant some participants rarely, if at all, engaged in exercise or were limited in how much or how often they could exercise. As one woman expressed, some did not view exercise as an important factor in their life:

*Working out is not. . . it's a luxury. It's not a privilege. It's a luxury. And, because of that, a lot of people don't work out. (P3)*

Living with a mental health condition, such as depression, made exercising difficult to achieve for some women. Lacking motivation to become active resulted in this participant avoiding exercise:

*I'm lazy and I need some motivation. . .. I have depression issues and it's kind of hard to snap out of and go think about going to work out. . . (P10)*

Alternatively, those who viewed exercise as an integral part of their life, made exercise a priority. Some participants expressed they were self-motivated, and refused to accept excuses for not exercising; irrespective of environmental or personal factors, they engaged with exercise to remain healthy. For instance, one woman stated:

*Like, you have time to check your email, you can have time to do some exercise if you want. Even if you have a kid, I saw millions of girls doing exercise with kid in the house if they wanted. . . if you really want to do, you-you-you can find it, you can find time. (P7)*

ii) *Exercise as a health promotion.* Regardless of their exercise status, women perceived exercise as important for health promotion that could prevent secondary disease with aging and multi-morbidity. Participants described some form of improvement, from breathing better or losing weight, to enhancing health when they engaged with exercise at some point in life:

*After I discovered, the-the benefits from movement and walking, in my personal growth in one year. I think [exercise] is extremely, extremely, extremely, extremely, extremely beneficial because it helps mental health and it also helps physical health. It helps you get alive again. (P9)*

Another example of exercise conceptualized as a health promotion strategy included:

*the reason of exercise and then the important as a women living with HIV, maybe aging also is to reduce some mental health issues. . . And then, they also reduce some risk of-of sickness. For example, diabetes, for example, heart attack, for example reduce cholesterol. So, all kind of important to do exercises that is most important for. (P2)*

iii) *Terminology.* Women's perceptions of exercise revealed that the majority had variable perceptions of the distinctions between 'exercise', as planned and structured and 'physical activity', as encompassing daily activities. For example:

*I didn't know the difference [between the terms], but uh I think it's good that there's a difference because there is. Because in exercise, you get other kinds of movements happen, right? In exercise, you have these movements that are essential for all the growth that I talked about earlier. . .. And, you get movement with physical activity too, but I think it's a different kind when you know you got the heart happening, this happening. Mind you, I do get it when I run up and down the stairs too. (P9)*

Participants had varying views on the differences between the term's 'exercise' and 'physical activity' and their respective meanings, for instance:

*I feel like exercise is more like voluntary, where you actually have to tell yourself you have to do this. But physical activity is where you're just doing it because you have to. (P5)*

Some women also considered themselves 'exercisers' even though they did not achieve the CSEP guidelines. This woman, for instance, after being informed of the terms, realized this during the interview: *just when I was thinking I was exercising, I wasn't really. (P10)*

However, participants stated that knowing the differences between the terms 'exercise' and 'physical' activity could help when recommending exercise for women living with HIV, and provide direction on how to achieve the physical activity guidelines

*Yes, it's good to know the term[s] and to know which is—what is because it's make more clearly for you what you need to do. (P7)*

**Facilitators and barriers to exercise.**   Facilitators and barriers included factors that influenced engagement with exercise among women living with HIV. Facilitators to exercise included: 1) aspirations to achieve a healthy lifestyle, 2) using exercise as a mental diversion from stressors in life, 3) having an exercise companion, 4) and receiving financial support from community-based organizations to facilitate engagement in exercise. Conversely, barriers to exercise included: 1) limited resources, such as lack of mental-health support and fitness resources, 2) financial limitations, 3) time and gym restrictions, and 4) cold winter weather conditions. See Table 3 for a description of the facilitators and barriers with supporting quotes.

**Strategies for uptake of exercise.**   Participants described personal and organizational strategies to facilitate the uptake and sustainability of exercise among women living with HIV. Personal strategies included: 1) creating social interactions through classes to help women maintain their exercise routine and maintain accountability by having their peers to exercise along with; and organizational strategies included: 2) provision of online exercise classes, 3) raising awareness and educating the population to make informed health decisions, and 4) offering practical support, such as child-care and financial support. See Table 4 for a description of the strategies with supporting quotes.

## Discussion

In this study, we explored the intersecting components that comprised experiences of exercise from the perspectives of women living with HIV in Canada. The six intersecting components include: (1) culture, (2) gender, (3) HIV-related stigma, (4) episodic nature of HIV, (5) sense of belonging, and (6) perceptions of exercise. While exploring the intersectionality of the six components was beyond the scope of this study, it is important to acknowledge that interconnecting patterns emerged between the various components. Intersectionality helps us understand that a woman living with HIV is not 'only' a woman, but can also possess other social identities that can intersect, for example: being a Black woman, a mother, a trans woman, a disabled woman, a woman with low socioeconomic status, or any other components of her identity that may generate empowerment or oppression [44–46]. Accordingly, an intersectional methodological approach can lead to multidimensional solutions that address the intricate conditions of women and their social identities [47].

Cultural roles and expectations articulated by ethnic minority women in this study were consistent with prior research that reported African American women are more subject to

Table 3. Facilitators and barriers to exercise among women living with HIV.

| Facilitators to Exercise | Description | Supporting Quotes |
|---|---|---|
| 1) Aspirations to achieve a healthy lifestyle | • Goals to become mentally and physically healthier and maintaining a healthy mental outlook on life. | ○ I do want to get uh like a stronger heart. I want to make sure that I keep maintaining my low blood pressure and my low cholesterol and eating right is something that I've managed to do. (P4)<br>○ So, what motivates me now to exercise is that I don't want to fall into a depressive hole again. (P9)<br>○ what other incentive do you have other than to get healthy. . . especially when you've had bad health. . . you have respect for good health. Health is wealth, you know? (P8) |
| 2) Using exercise as a diversion from stressors in life. | • Using exercise as a distraction from stressful issues that may be impeding one's life. | ○ Well, if I'm going through something, exercise helps me deal with issues that I'm going through in life. . . I'm able to vent out my emotions through exercise. (P6)<br>○ If you watch TV or watch the phone or you cook something, you may—the mind is may go to some problem. But, when you do exercise, if you are really focusing about it. You focus on that so you don't have time to think for something else. Like, before maybe I go to sleep when I think, "oh this, that, that" you know, so many thoughts. But, now because the body's tired, you sleep right away at 10–11 when you go to sleep, you don't stay in the bad thinking of problems or something like that. (P7)<br>○ I still reminded that I have uh, that, I-yeah "yeah, I may die." Yeah, before everyone else because I am HIV-positive. So, then I worry about the future of my family, I worry about the future of myself because I don't have someone in my family who take care of me. . . So, exercising will uh, help me to, so when I get out, when I'm about to go out, I yeah, I focus on what I'm doing not worrying. Like I get distracted and then uh, when I get tired, I come and sleep. I also have that feeling that I accomplished something, so which will take me those worries. . . exercising will help with not much time for worries. (P1)<br>○ usually if I-if I am in poor mind state, working out will actually help me. . . (P3)<br>○ I find that whenever you exercise, you know, you're not that stressed about as necessarily about too many things. . . I find myself more relaxed. (P5) |
| 3) Having an exercise companion | • Women felt more inclined to exercise when they had encouragement from peers. | ○ you make uh friends and then they encourage you will come back and you will say "I will be there next week, will you come with me?" You go again and you go again and you go again. And then you're active. . . (P2)<br>○ if other friends are participating, I'll go and do something like that. But, I'm not the type of person to engage on my own. (P4)<br>○ it's great when you have somebody working out with you, makes it more fun, right? Like, a buddy for like if someone's having a bad day then your friend that you usually work out with can give you a boost, you know? If you're not feeling up to par, right? (P6) |
| 4) Financial support from community-based organizations | • Financial support in the form of discounted gym memberships from various HIV community-based organizations to join or access a gym facility. | ○ I'm lucky to have the, uh, do you know the uh- [name of CBO]. Yeah, I get 50% of what I pay for exercise. So, yeah, what I do is pay like uh 6 months and then I send, yeah, then I give the receipt to [CBO] and they will have a half, half of it back. . . they-they pay 50%. . . very helpful. (P1)<br>○ I have no problems because, you know, when I get my membership for the gym, uh [CBO] gives back half of my money so if it costs me $300 for a membership, it's only costing me $150 'cuz I get $150 back. . . So, uh there's resources that I can reach out to and take advantage of, being uh HIV-positive. (P6) |
| Barriers to Exercise | Description | Supporting Quotes |

(*Continued*)

**Table 3.** (Continued)

| | | |
|---|---|---|
| 1) Limited resources (including lack of mental-health support and exercise classes or groups) | • Shortage of mental-health resources within the HIV community that could otherwise help facilitate engagement with exercise by allowing women to overcome psychological barriers.<br>• Limited exercise classes or groups for women living with HIV, and sub-groups of women living with HIV. | ○ *There's not enough mental health practices in our community... we don't have a lot of support.... The only way you can make this meaningful for women is to get a psychiatrist on board so that they can be treated while they are exercising because no amount of exercising is going to help you if you don't have a good psychiatrist... You need, people who get it, who understand the well-being and the psych, and the micro-aggressions that go on with women of color.... that's a huge barrier with why women don't work out... the only way you're going to switch the narrative of women getting healthier is getting their mind right. (P3)*<br>○ *That's why psychotherapists, you shouldn't have to pay for them. They should be covered or you know, included in OHIP ... Yeah [having like a psychotherapist or a therapy support in addition to exercise will optimize women's health conditions]. Yes, and not a psychiatrist though because a psychiatrist is just like a doctor, he's going to give you another pill to fix your mental health stuff. (P8)*<br>○ *we have some organization, which I am registered, you know, specific to this diagnosis and they have class for food, class for medication, class like no class, meetings, you know, sessions like, "this Tuesday we talk about for—" but never say, "you know what, take a t-shirt and follow me, guys, do this do that for 10 minutes." You know, because this laziness and comfortable things for people to sit and just watch... (P7)*<br>○ *There's not something that says, 'hey women living with HIV come!' its 'Black women, Caribbean women living with HIV come and talk to us.' So, that was about the only challenge... I just gave up. (P4)* |
| 2) Financial limitations | • Lack of finances for a gym membership or exercise class. | ○ *I am on ODSP. I also under the table work a couple of jobs, uh but I can't afford it. What I-what I make and what I have is just barely affords me, you know, food to eat and uh a couple of bad habits that I have and, you know, like transportation stuff like that, that's all that I can afford. I have no extra like- it's crazy the amount of money the Y took for joining the gym.... (P8)*<br>○ *I don't have money to join a gym. (P10)*<br>○ *No, [I have never taken an exercise class] because everything was paid so I didn't really have the money to pay for-for the class. To go, yeah, to go for membership or stuff like that. (P7)* |
| 3) Time and gym restrictions | • Limited exercise engagement due to employment, family responsibilities or other life priorities.<br>• Restricted hours for gym operations made it difficult to join a gym or fitness class. | ○ *I wake up at 4 am in the mornings [for work], so this is like, my 9 o'clock at night. So, I think I got a short window...The times that I go, like what classes are available at my time for work? (P4)*<br>○ *it's nice to go to the gym, but to work out at home is-is great too... 'cuz you can work out at any time. I find at gyms... you go to a gym, you're too restricted to like I don't know, 9 to 9 or something like that? Uh, having to wait sometimes for-for different machines or different uh however methods you want to use to train that sometimes you're waiting for equipment. (P8)* |
| 4) Cold winter weather conditions | • Environmental weather barriers (cold, winter temperatures and allergies) make it challenging to exercise. | ○ *I'm allergic to the cold so I do not exercise in the winter outdoors... I break out in hives... or air conditioning and getting warm again so I always found that way is challenging...I have to keep my body very regulated.... So, if I'm in an environment where I get too hot in that environment, then it starts to get cold then hot again... (P4)*<br>○ *winter, the cold weather can be a barrier [to exercise] .... (P1)* |

unhealthy behaviours, including lower levels of physical activity [48, 49] Moreover, racial or ethnic minority women living with HIV endure high levels of psychological distress [50] possibly due to the intersection of various social identities they embody [51–53]. Experiences of racism may also contribute to HIV-related stigma in women [54], with marginalized groups facing several forms of discrimination (i.e., racism, sexism, poverty) that can exacerbate HIV-

**Table 4. Strategies for engaging in exercise among women living with HIV.**

| Strategy | Description | Supporting Quotes |
|---|---|---|
| 1) Create social interactions | • Creating social interactions (i.e., exercise classes or a person to exercise with) may encourage accountability to an exercise regimen.<br>• Social interactions generate a supportive and encouraging atmosphere to exercise. | *Maybe offering uh training mentor, training buddies. . . once they lift the ban and they—and the restrictions, really encouraging women, you know, we have a lot of support groups. Maybe having uh a support group that just exercised. That's it. Like, you know there's [CBO], they have wonderful space for women who want to work out in a safe space who don't want to go to a gym. Hire a trainer and have those women go 3 times a week. They have coffee night twice a week. Have them go to [CBO], like there's tons of s— or even go out here in City Place. There are a ton of places for women to feel safe around their own gender. . . around their own ethnicity. . . take it out of the gym, take it off of zoom and make it more connection-connected. (P3)*<br>*Accountability. Like, having someone rely on me to just be like, if I'm not meeting with them, then I'm wasting their day. . . you gotta be able to-to put yourself out there and because then somebody else is relying on me. And, I'm such that type of person to be one of the caregiver and the things like that, that I wouldn't let them down. . . just having somebody to just get into your back. (P4)*<br>*I need to get somebody who is on board with me to uh—for me and some of my friends to inspire each other to exercise, to make the time and, you know, do something about it. . . (P10)*<br>*First of all, there are people, you go outside and you have some place to go and come back. You dress a little bit, you know, get a nice t-shirt some shorts. You go there, see the other people. . . it's helpful, you know, then-then do it by yourself in the house. So, this social things is really good too. (P7)*<br>*What can I to motivate is just like uh we meet all the time like to encourage me, to remind me " did you do this?" . . . Care is also something very important to make, to make motivate somebody. . . It can encourage me. (P2)* |
| 2) Provision of online exercise classes | • Convenience and accessibility of online classes to diverse lifestyles.<br>• Preferences for sustained online exercise programming post COVID-19 due to accessibility.<br>• Flexibility to exercise in the comfort of own home and shifting control to those who prefer to work out on their own time. | *Something like [online classes] would be beneficial to start your— especially since COVID's still here. And, even if-even if gyms and all that are opening up and whatever, for a lot of people it might not be the right thing to do right now. So, doing online stuff is the best thing you could probably do for some- the people from home, or evening, or weekends or parts that work when if-if those people are working. Now, in my case, my work is no pay work, it's a different kind of work. But, you know, some people have a 9–5 work, so yeah, something weekends or whatever, an online thing might work. (P9)*<br>*I find that the online groups right now are helping women to get motivated. So, I think even with everything starting to open up again, I still think that somethings should still remain online. . .. because they're-they're in their own space so they don't—and then it opens their mind that they don't have nobody around them to distract them. And, then it makes them get involved. . . (P6)*<br>*I like the uh online thing. I think that's the best thing for me. I don't feel like I need to go to a gym. . . (P8)* |

(*Continued*)

**Table 4.** (Continued)

| Strategy | Description | Supporting Quotes |
|---|---|---|
| 3) Raising awareness and educating the population to make informed health decisions | • Promoting uptake through provision of educational resources on HIV and exercise.<br>• Encouraging community members, leaders and health care providers to discuss various benefits of exercise and how to integrate exercise into daily life.<br>• Placing educational pamphlets and other means of exercise promotion in medical offices since women are frequently visiting their health care providers. | *when you invite the women for like a yoga session or Zumba or whatever it is, you have to talk about the importance of doing those things consistently... There has to be somebody emphasizing the importance of doing [exercise]. Even it can be something that can be done before the exercise starts. Just a quick reminder... of the importance of exercise in your life. (P5)*<br>*to promote [exercise], I think uh, having more topics in the women's group because they have for women who are comfortable joining the groups, different groups like inviting someone to talk about the benefit of exercising in those groups. So, that will be helpful... also involve the family, the-their health-care providers... yeah, I think they do so because my family doctor and my specialist always do tell me to exercise too. (P1)*<br>*when I go to my doctor, she asks me, "did you go to dentist? Do you sweat? Do you drink water? Do you smoke? Do you do this?" She never asks me, "do you exercise? Do you know where to go to exercise?" Never. Nothing for years. So, that's a good way to talk to the people, straight away tell them. You as a doctor, as a specialist, tell them, "look there is this place, there is this way, we have this flyer for you" or like you say this meeting, this organization, all the time repeat sure! Because they repeat about this, "did you take your vitamins? Did you take your medication?" But, never say, "did you do you exercise today?" So, should be a constant reminder to the people. (P7)*<br>*Putting [resources] out there in different places. My doctor's office has all these pamphlets, but it's 'men engaging in this,' 'men doing this,' 'men doing that.' Having it there—I mean there's one pamphlet that I saw was 'pregnancy and HIV.' Not all of us are getting pregnant. Like, so having it somewhere where we go, that we need to go. So, your doctor's office, those kind of things where you do go and clinics and things that you do... I think having it there, when you're looking and just highly being noticed on a freakin' wall of pamphlets, that you know—I mean, I'm sick of seeing breast cancer, but you know, I gotta go look at 'talk to man' studies if I want to get anything to do with, you know, my condition. (P4)* |
| 4) Practical support | • Offering practical support from community-based organizations such as a) child-care and b) financial support (i.e., free gym memberships) to allow women to focus on their fitness goals. | *You know, offer to pay for their child care so that they can come and work out. (P3)*<br>*I'd like to see [women] being able to access it easily, you know, with the child-care if needed and being motivated with uh classes, sure. And-and, free memberships, that'd be great (P10)*<br>*I think especially women need money. So, I think that there's a financial sort of like an honorarium... maybe if they started with an honorarium that the exercise, uh the momentum might just keep up and you don't have to offer an honorarium after that... child care is another thing for a lot of women... because maybe they don't want to take their kids to the gym or while they go workout ... (P8)*<br>*there are barriers like taking care of the family members... someone who has children, let's say they have a group to meet other women and that way they uh, the organizer can have the budget for who stays with children.... (P1)*<br>*if they can be something to help with the children... even just like uh babysitting ... I don't know if that could be done by families or organizations... if you can say, "you know what? Every-every day, at this time to this time, there's babysitting available for you, you go and do your exercise. (P5)*<br>*[Organizations] should make a way to make some free gym pass for some people (P7)* |

related stigma [51, 53, 55]. Accordingly, an intersectional framework offers a holistic approach on understanding various stigmatized identities (i.e., racialized women living with HIV) and its effects on health outcomes [56–59] and perhaps how this may hinder women's experiences with exercise.

Compared to their HIV-negative counterparts, women living with HIV are often more impacted by traditional gender norms [60], and specific social gender-related roles such as caregiving can impact self-management strategies in women living with HIV [61]. A commonly agreed upon benefit of exercise among women living with HIV also included improving their body physique [62]. Several women in this study expressed wanting to exercise in order to "look good" by losing weight and toning their body due to lipodystrophy and atrophy attributed to HIV or associated treatments. To address these issues, offering flexible options for exercise such as online forms of community-based exercise may thus allow women to overcome barriers to exercise by engaging in exercise from the comfort of their own home.

Further, HIV-related stigma can negatively determine one's health with HIV [63, 64]. Perceived risks including fear of contamination or spread of germs are barriers to exercise for people living with HIV [41]. Fear of judgement by others, as mentioned by some study participants about their fellow community members, was a barrier to exercise. This fear is reflective of HIV-related stigma [65, 66]. This also mirrors the recent work of Vancampfort et al. (2021), who revealed that possessing higher levels of internalized HIV-related stigma was linked with lower levels of physical activity in people living with HIV [67]. Stigma may be experienced differently by gender as women living with HIV experienced higher rates of HIV-related stigma compared to men living with HIV [68, 69]. Researchers should further investigate the ways in which interventions on HIV stigma-reduction may enhance participation in exercise and improve health outcomes for women living with HIV [67].

With HIV being episodic in nature, it involves changing levels of health, which influences people's willingness to exercise [70]. Physical barriers to exercise among people living with HIV are characterized as side effects to HIV and HIV medication, which resulted in lipodystrophy and fatigue, breathlessness, muscular and joint pain and other conditions related to their co-morbidities [41, 43, 62, 71]. Such health fluctuations can generate uncertainty for people living with HIV and consequently make it challenging to establish health-promoting behaviors such as exercise [70]. However, Gielen et al. (2001) found that women living with HIV who practiced health promoting behaviors like exercise described improved health outcomes and quality of life. [72] As such, exercise can help avert and mitigate the acute effects of chronic comorbidities [73–77], which means that the health benefits of exercise are necessary to examine in future exercise-based interventions [78]. Evidence also supports the role of health care providers to ensure exercise is a priority against chronic conditions [79] as educating women on the benefits of exercise can potentially facilitate activity [80].

As exhibited within this study, social support is a key contributor in fostering exercise and preventing barriers to exercise in people living with HIV [81], and women living with HIV who possess a strong social support system describe improvements in both their health and quality of life, as well as facilitated their engagement with exercise [72, 82]. Social support is well evidenced as a facilitator for exercise and self-management strategy among people living with HIV [65, 81, 83, 84], whereas insufficient social engagement appears to negatively affect participation in exercise among people living with HIV [85]. Similarly, Peterson (2010) stressed the importance of maintaining supportive resources as some women living with HIV did not have a community network they could depend on [86]. This further suggests that social support can positively affect adherence to exercise [87], and offer a source of empowerment and sense of belonging to women living with HIV [88, 89]. As such, social support is a positive and effective addition to traditional HIV treatment methods [90–92].

Finally, similar to Simonik et al. (2016), participants in this study unveiled that their willingness to exercise was subjective to the perceptions they held about exercise [70]. While women acknowledged the importance of incorporating exercise into their life, many regarded it as a low priority [70]. 'Non-exercising' women may consider themselves as active individuals since their perception of exercise may be largely centered on social context [93, 94]. Findings from this study indicated that some 'non-exercising' women were unaware they were considered a non-exerciser on the basis of the CSEP guidelines. It appeared women classified as 'non-exercisers' had lower expectations for exercise when viewed as a construct [93]. Providers and researchers should further study women living with HIV and their perceptions of exercise and physical activity to confirm exercise interventions meet their unique needs [95].

## Strengths and limitations

This study presents several limitations. Purposive sampling allowed us to explore diverse experiences and perceptions of women who engaged or did not engage in exercise. Recruitment challenges resulted in a smaller than anticipated sample size, potentially attributed to fatigue with online meetings that took place during the COVID-19 pandemic [96] and other competing priorities of women. Nevertheless, our sample of racially diverse women was representative of the larger population of women living with HIV enhancing transferability of study findings [97]. Finally, this study was specific to women living in Toronto, a large urban centre in a high-income country, with access to technology, making it difficult to transfer finding with women living with HIV in rural areas with limited access to technology.

## Conclusions

Experiences with exercise among women living with HIV were characterized by six intersecting components: gender, culture, episodic nature of HIV, HIV-related stigma, sense of belonging and perceptions on exercise. Facilitators to exercise included: aspirations to achieve a healthy lifestyle; using exercise as a mental diversion from stressors in life; having an exercise companion; and receiving financial support from community-based organizations. Barriers to exercise included: limited resources; financial limitations, time and gym restrictions; and winter weather conditions. Strategies to facilitate the uptake and sustainability of exercise, included: creating social interactions through classes to help women maintain their exercise routine and maintain accountability by having their peers to exercise along with; provision of online exercise classes; raising awareness and educating the population to make informed health decisions; and offering practical support, such as child-care and financial support. Results from this study will help to inform the future tailored implementation of exercise as a rehabilitation strategy for women living with HIV in the community.

## Supporting information

**S1 File. REB approval letter.**
(PDF)

**S2 File. Interview guide.**
(PDF)

**S3 File. Details of exercise engagement (type, frequency and duration of exercise) among women living with HIV based on interview data.**
(PDF)

## Acknowledgments

We are deeply grateful to the women who participated in this study because without them this research would not be possible. This research was completed in fulfilment of the requirements for a Masters in Science degree from with the Institute of Health Policy, Management and Evaluation (IHPME), Dalla Lana School of Public Health, University of Toronto. We thank Casey House, AIDS Committee of Toronto, and Maple Leaf Medical Clinic for their support with recruitment. Thank you to community expert, Stephanie Smith, for engaging in a mock interview with the interviewer and provision of feedback on the interview guide and demographic questionnaire. Finally, we acknowledge Dr. Francisco Ibáñez-Carrasco for facilitating many knowledge translation opportunities along the way.

## Author Contributions

**Conceptualization:** Nora Sahel-Gozin, Kelly K. O'Brien.

**Data curation:** Nora Sahel-Gozin.

**Formal analysis:** Nora Sahel-Gozin.

**Funding acquisition:** Kelly K. O'Brien.

**Investigation:** Nora Sahel-Gozin.

**Methodology:** Nora Sahel-Gozin.

**Resources:** Kelly K. O'Brien.

**Supervision:** Mona Loutfy, Kelly K. O'Brien.

**Validation:** Mona Loutfy, Kelly K. O'Brien.

**Writing – original draft:** Nora Sahel-Gozin.

**Writing – review & editing:** Mona Loutfy, Kelly K. O'Brien.

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
