## [Decision Letter · Decision Letter 0]

29 Mar 2023

PONE-D-22-32552Experiences Engaging in Exercise from the Perspectives of Women Living with HIV: A Qualitative StudyPLOS ONE

Dear Dr. O'Brien,

Thank you for submitting your manuscript to PLOS ONE. After careful consideration, we feel that it has merit but does not fully meet PLOS ONE’s publication criteria as it currently stands. Therefore, we invite you to submit a revised version of the manuscript that addresses the points raised during the review process.

We look forward to receiving your revised manuscript.

Kind regards,

Enock Madalitso Chisati, PhD

Academic Editor

PLOS ONE

Journal Requirements:

Reviewers' comments:

Reviewer's Responses to Questions

**Comments to the Author**

1. Is the manuscript technically sound, and do the data support the conclusions?

Reviewer #1: Yes

2. Has the statistical analysis been performed appropriately and rigorously? 

Reviewer #1: N/A

3. Have the authors made all data underlying the findings in their manuscript fully available?

Reviewer #1: Yes

4. Is the manuscript presented in an intelligible fashion and written in standard English?

Reviewer #1: No

5. Review Comments to the Author

Reviewer #1: This paper reports findings of a descriptive study designed to explore experiences and perspectives regarding the culture of habit of exercising. The study population are women living with HIV (WLWH). The study adopts a qualitative research design with purposive sampling, uses a combination of structured interviews and questionnaires, and analyses results mainly through thematic analysis. As such statistical rigor was not a necessity

One of the great strengths of this papers is its possession of a rigorous presentation of participants’ demographic characteristics, levels of exercise engagement, and factors (culture, gender, stigma, etc.) that affect exercise engagement among WLWH which serve to illuminate underlying drivers and barriers guiding adoption as well as continuance of a pro-exercise culture in this population group. The article moves further to make an important claim regarding intersectionality of these component factors. However, it does not sufficiently demonstrate how these components of an experience are connected. The authors suggest that this may be a scope for future studies, which is agreeable.

All in all, the manuscript is technically sound, data collection methods were scientifically plausible, and results (though not discussed in-depth) offer not a lot of insights that are sufficiently informed by the data.

Therefore, this paper may be accepted after careful consideration of issues laid out below:

1. While the result section of the paper is rich and informative, sentence construction needs to be revisited in multiple areas to improve readability. Specifically, authors need to make sure that incorporation of numbering within a paragraph does not distort the structure or message of the sentence. (e.g. lines, 403-408, 453-457, 597-583, 519)

2. I understand the likely need to include participants’ quotes verbatim. However, the structuring of some of these original quotes to display hesitancies, false starts, and pauses which participants made during interviews, makes the arguments difficult to follow. Moreover, the presentation of emotional context and nuances of regional dialect may be a disservice to participants and the research in general by potentially compelling readers to associate certain ideas or lines of thought with ethnicity, which may be unintended. Therefore, I advise the authors to edit and truncate such quotes where possible (including omitting offensive terms (page 29, table 3) to improve readability and objectivity. Works by Corden & Sainsbury (https://www.york.ac.uk/inst/spru/pubs/pdf/verbquotresearch.pdf) present best arguments on this activity.

3. Lastly, the authors should revisit or re-classify the groupings in relation to level of exercise or physical activity engagement (table 2 & lines 210-214) to ensure that groups are mutually exclusive. This is to avoid confusion on those points.

6. PLOS authors have the option to publish the peer review history of their article (what does this mean?). If published, this will include your full peer review and any attached files.

Reviewer #1: **Yes: **Charles Nyasa

---

## [Author Response · Author response to Decision Letter 0]

8 May 2023

Reviewer Comments

1. Comment: While the result section of the paper is rich and informative, sentence construction needs to be revisited in multiple areas to improve readability. Specifically, authors need to make sure that incorporation of numbering within a paragraph does not distort the structure or message of the sentence. (e.g. lines, 403-408, 453-457, 597-583, 519)

Response: Thank you for your comment. We have carefully revised the manuscript to ensure that sentence structure is clear and readable. Further, we believe that the incorporation of numbering within a paragraph (i.e., page 12, lines 233-235 – clean version) is to provide organization and allow readers to easily visualize the topics that will be discussed in the text. Please view tracked changes for specific edits. 

2. Comment: I understand the likely need to include participants’ quotes verbatim. However, the structuring of some of these original quotes to display hesitancies, false starts, and pauses which participants made during interviews, makes the arguments difficult to follow. Moreover, the presentation of emotional context and nuances of regional dialect may be a disservice to participants and the research in general by potentially compelling readers to associate certain ideas or lines of thought with ethnicity, which may be unintended. Therefore, I advise the authors to edit and truncate such quotes where possible (including omitting offensive terms (page 29, table 3) to improve readability and objectivity. Works by Corden & Sainsbury (https://www.york.ac.uk/inst/spru/pubs/pdf/verbquotresearch.pdf) present best arguments on this activity.

Response: Thank you for the resource. We removed as many hesitences, false starts, and pauses from the original quotes as possible without disrupting the integrity and meaning of the participant’s quote. We also attempted to eliminate as many quotes presenting emotional context and nuances of regional dialect as possible. Overall, we were careful with how much text and characters we removed from the original quotes as our goal is to retain the true meaning of quotes presented by participants, which adds to this study’s richness. Please view tracked changes for specific edits.

3. Comment: Lastly, the authors should revisit or re-classify the groupings in relation to level of exercise or physical activity engagement (table 2 & lines 210-214) to ensure that groups are mutually exclusive. This is to avoid confusion on those points.

Response: Thank you. In this study, we report on exercise engagement as determined by a self-reported questionnaire and interview data (page 10-11; lines 210-223), which are both mututally exclusive. Thus, as noted in the methods section, participants were categorized as an ‘exerciser’ or ‘non-exerciser’ using a combination of quantitative and qualitative approaches, including responses to the demographic questionnaire and their interview data. Participants who were categorized as non-exercisers based on their interview data were further categorized into one of the following three options: 1) currently does not exercise, 2) currently does exercise but does not meet the CSEP guidelines, or 3) currently does not exercise but was meeting the CSEP physical activity guidelines at some point in the past. Table 2 demonstrates participants’ level of physical activity as they reported on the demographic questionnaire. Additionally, data collected from each woman during their interviews also allowed us to better understand their engagement levels with exercise as we asked several questions regarding the nature and extent of their exercise (objective 1 of the study- see S2 file interview guide). Thus, we used two methods of data collection to obtain women’s exercise and physical activity status: interview data and demographic questionnaire. As noted on page 37 lines 749-758 (clean version), each method yielded different responses where women largely viewed themselves as exercisers (as exhibited in the demographic questionnaire), however, based on the narrative in their interviews, they did not actually meet the CSEP guidelines.

---

## [Editor Report · Decision Letter 1]

18 May 2023

Exploring Experiences Engaging in Exercise from the Perspectives of Women Living with HIV: A Qualitative Study

PONE-D-22-32552R1

Dear Dr. O'Brien,

We’re pleased to inform you that your manuscript has been judged scientifically suitable for publication and will be formally accepted for publication once it meets all outstanding technical requirements.

Kind regards,

Enock Madalitso Chisati, PhD

Academic Editor

PLOS ONE
---

## [Editor Report · Acceptance letter]

23 May 2023

PONE-D-22-32552R1 

Exploring Experiences Engaging in Exercise from the Perspectives of Women Living with HIV: A Qualitative Study 

Dear Dr. O'Brien:

I'm pleased to inform you that your manuscript has been deemed suitable for publication in PLOS ONE. Congratulations! Your manuscript is now with our production department. 

Kind regards, 

on behalf of

Dr. Enock Madalitso Chisati 

Academic Editor

PLOS ONE